# Decoupled Self-supervised Learning for Graphs

**Teng Xiao[1], Zhengyu Chen[2], Zhimeng Guo[1], Zeyang Zhuang[3], Suhang Wang[1]**
[1]The Pennsylvania State University, [2]Zhejiang University, [3]Tongji University
{tengxiao,zhimeng,szw494}@psu.edu,chenzhengyu@zju.edu.cn
zeyangzhuang0315@gmail.com

## Abstract

This paper studies the problem of conducting self-supervised learning for node representation learning on graphs. Most existing self-supervised learning methods assume the graph is homophilous, where linked nodes often belong to the same class or have similar features. However, such assumptions of homophily do not always hold in real-world graphs. We address this problem by developing a decoupled self-supervised learning (DSSL) framework for graph neural networks. DSSL imitates a generative process of nodes and links from latent variable modeling of the semantic structure, which decouples different underlying semantics between different neighborhoods into the self-supervised learning process. Our DSSL framework is agnostic to the encoders and does not need prefabricated augmentations, thus is flexible to different graphs. To effectively optimize the framework, we derive the evidence lower bound of the self-supervised objective and develop a scalable training algorithm with variational inference. We provide a theoretical analysis to justify that DSSL enjoys the better downstream performance. Extensive experiments on various types of graph benchmarks demonstrate that our proposed framework can achieve better performance compared with competitive baselines.

## 1 Introduction

Graph-structured data is ubiquitous in the real world, such as social networks, knowledge graphs, and molecular structures. In recent years, graph neural networks (GNNs) [16, 27, 52, 7, 60] have been proven to be powerful in node representation learning over graph-structured data. Typically, GNNs are trained with annotated labeled data in a supervised manner. However, collecting labeled data is expensive and impractical in many applications, especially for those requiring domain knowledge, such as medicine and chemistry [71, 20]. Moreover, supervised learning may suffer from problems of less-transferrable, over-fitting, and poor generalization when the task labels are scarce [10, 64].

Recently, self-supervised learning (SSL) provides a promising learning paradigm that reduces the dependence on manual labels in the image domain [5, 12, 13, 6]. Compared to image data, there are unique challenges in designing self-supervised learning schemes for graph-structured data since nodes in the graph are correlated with each other rather than completely independent, and geometric structures are essential and heavily impact the performance in downstream tasks [27]. A number of recent works [53, 18, 64, 69, 48, 65, 45] have studied graph self-supervised learning and confirm that it can learn transferrable and generalizable node representations without any labels. Typically, there are two main self-supervised schemes to capture structure information in graphs [53, 64, 45]. The first scheme is reconstructing the vertex adjacency following traditional network-embedding methods [26, 14, 17, 16, 46], which learns an encoder that imposes the topological closeness of nodes in the graph structure on latent representations. The key assumption behind this scheme is that neighboring nodes have similar representations [53, 46]. However, this assumption over-emphasizes proximity [53, 64, 47] and does not always hold true for heterophilic and non-homophilous (mixing) graphs. In comparison, contrastive learning methods [53, 18, 64, 69, 48, 65] construct two graph

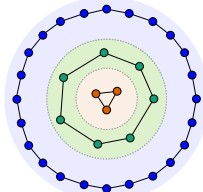 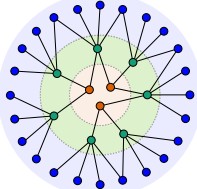 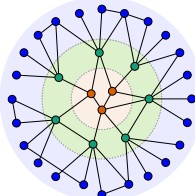

(a) homophilous pattern.   (b) heterophilic pattern.   (c) mixing pattern.

Figure 1: An illustration examples of different types of graphs. Nodes with similar labels typically have similar neighborhood patterns in all types of graph. This assumption is general than the standard homophily assumption where nodes with similar semantic label typically be linked with each other.

views via the stochastic augmentation and then learns representations by contrasting views with information maximization principle. While these contrastive methods can capture structure information without directly emphasizing proximity, their performance relies on topology augmentation [70, 50]. Importantly, conducting augmentation for non-homophilous graphs is relatively difficult since linked nodes may be dissimilar, and nodes with high similarities might be farther away from each other. Hence, the above problems pose an important and challenging research question: *How to design an effective self-supervised scheme for node representation learning in non-homophilous graphs?*

We approach this question by taking advantage of neighborhood strategies of nodes for the self-supervised learning on non-homophilous graphs. Our key motivation is that nodes with similar neighborhood patterns should have similar representations. In other words, we expect that the neighborhood distributions can be exploited to distinguish node representations. Our assumption is more general than the standard homophily assumption as shown in Figure 1. For instance, while the gender prediction in common dating networks lacks homophily [1], neighborhood distribution is very informative to the node gender labels, i.e., nodes with similar neighborhoods are likely to be similar.

While the motivation is straightforward, we are faced with two main challenges. The first challenge is capturing the local neighborhood distribution in a self-supervised learning manner. The neighborhood distributions typically follow heterogeneous and diverse patterns. For instance, a node usually connects with others due to the complex interaction of different latent factors and therefore possesses distribution consisting of local mixing patterns wherein certain parts of the neighborhood are homophilous while others are non-homophilous. The lack of supervision obstructs us from modeling the distribution of neighborhoods. The second challenge is capturing the long-range semantic dependencies. As shown in Figure 1, in non-homophilous graphs, nodes with high semantic and local structural similarities might be farther away from each other. For this reason, global semantic information is the objective that we would incorporate for self-supervised learning.

This paper presents a new self-supervised framework, decoupled self-supervised learning (DSSL), to achieve a good balance between these two challenges. At the core of DSSL is the latent variables, which empower the model with the flexibility to decouple the heterogeneous and diverse patterns in local neighborhood distributions and capture the global semantic dependencies in a coherent and synergistic framework. Our contributions can be summarized as follows: (1) We propose a DSSL for performing self-supervised learning on non-homophilous graphs, which can leverage both useful local structure and global semantic information. (2) We develop an efficient training algorithm based on variational inference to simultaneously infer the latent variables and learn encoder parameters. (3) We analyze the properties of DSSL and theoretically show that the learned representations can achieve better downstream performance. (4) We conduct experiments on real-world non-homophilous graphs, and the results demonstrate the effectiveness of our self-supervised learning framework.

## 2 Related Work

**Non-homophilous Graphs.** Non-homophilous is known in many settings such as online transaction networks [36], dating networks [1] and molecular networks [68]. Recently, various GNNs [37, 68, 59, 29, 69, 8, 44, 62] have been proposed to deal with non-homophilous graphs with different methods such as potential neighbor discovery [37, 23, 22], adaptive message propagation [8, 59], exploring high-frequency signals [2] and higher-order message passing [68, 5]. Despite their success, they typically consider the semi-supervised setting and are trained with task-specific labeled data, while in

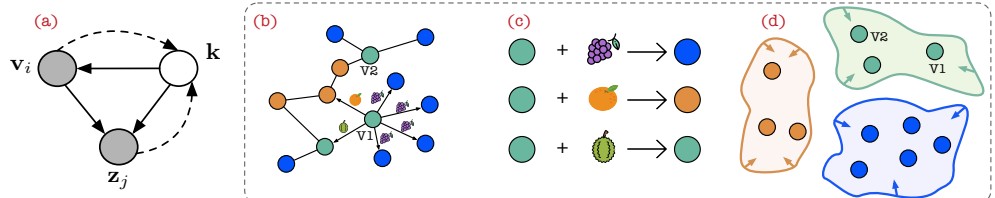

Figure 2: An illustration of (a) a graphical model for DSSL. The discrete latent variable $k$ is used to instantiate a latent mixture-of-Gaussians $\mathbf{v}_i$, which is then decoded to $\mathbf{z}_j$ and (b) a toy subgraph example of non-homophilous graphs where color denotes the class of the node. (c) Our model encodes the semantic shift by decoupling structure since different each node has different latent factors to make connections to its different neighbor, and (d) captures global semantic structure via the semantic clusters, which seeks to push the representations of $v_1$ and $v_2$ to their closest prototypes.

practice, labels are often limited, expensive, and even inaccessible. In contrast, in this paper, we study the problem of self-supervised learning: learning node representations without relying on labels.

**Self-supervised Learning on Graphs.** Self-supervised learning holds great promise for improving representations when labeled data are scarce [5, 12, 13, 19]. Earlier combinations of GNNs and self-supervised learning involve GraphSAGE [16], VGAE [26], and Graphite [15], which typically follow the traditional network-embedding methods [38, 46, 14] and adopt the link reconstruction or random walk principle. Since these methods over-emphasize node proximity at the cost of structural information [53, 45, 64], various graph contrastive learning methods have been proposed [53, 18, 64, 69, 48, 35], which aim to learn representations by contrasting representation under differently augmented views and have achieved promising performance. However, they heavily rely on complex data- or task-specific augmentations, which are prohibitively difficult for non-homophilous graphs. Our work differs from the above methods and aims to answer the question of how to design an effective self-supervised learning scheme for non-homophilous graphs.

**Disentangled graph learning.** Our work is also related to but different from existing disentangled graph learning that aims to decouple latent factors in the graph. vGraph [43] considers a mixture process to define the conditional probability on each edge. vGraph is a classical shallow network embedding algorithm. However, we instead tackle the problem of self-supervised learning with GNNs. There are a couple of works that explore the disentangled factors in the node-level [33, 32], edge-level [67] and graph-level [63]. Whereas these methods require task-specific labels that can be extremely scarce. By contrast, we address the problem of learning node representations on non-homophilous graphs without labels. Disentangled contrastive learning [28] learns disentangled graph representations without labeled graphs. [61] proposes the self-supervised graph-level representation learning with disentangled local and global structure [61]. Nevertheless, their goal is to conduct graph-level classification tasks, but we tackle a node-level representation problem. [66, 9, 55] consider adding a clustering layer or a prototype clustering component to capture the global information. However, they are still based on contrastive learning [66, 9] or supervised learning [55]. By contrast, our framework is an unsupervised generative model and does not rely on graph augmentations and downstream labels. Moreover, we focus on non-homophilous graphs where connected nodes may not be similar to each other by decoupling the local diverse neighborhood context.

## 3 Decoupled Self-supervised Learning

In this section, we describe our problem setting with respect to self-supervised learning and demonstrate our approach. An illustration of generative and inference processes is depicted in Figure 2 (a).

### 3.1 Problem Formulation

We consider a graph $G = (\mathcal{V}, \mathcal{E})$, where $\mathcal{V}$ is a set of $|\mathcal{V}| = N$ nodes and $\mathcal{E} \subseteq \mathcal{V} \times \mathcal{V}$ is a set of $|\mathcal{E}|$ edges between nodes. $\mathbf{A} \in \{0, 1\}^{N \times N}$ is the adjacency matrix of $G$. The $(i, j)$-th element $\mathbf{A}_{ij} = 1$ if there exists an edge $(v_i, v_j)$ between node $v_i$ and $v_j$, otherwise $\mathbf{A}_{ij} = 0$. The matrix $\mathbf{X} \in \mathbb{R}^{N \times D}$ describes node features. The $i$-th row of $\mathbf{X}$, i.e., $\mathbf{x}_i$, is the feature of node $v_i$. Given the graph $\mathcal{G} = (\mathbf{X}, \mathbf{A})$, the objective of self-supervised node representation learning is to learn an encoder

function $f_\theta(\mathbf{X}, \mathbf{A}) : \mathbb{R}^{N \times N} \times \mathbb{R}^{N \times D} \to \mathbb{R}^{N \times D'}$ where $\theta$ denotes its parameters, such that the representation of node $v_i$: $f_\theta(\mathbf{X}, \mathbf{A})[i]$, can be used for downstream tasks such as node classification.

## 3.2 The Probabilistic Framework

In this section, we introduce our framework, decoupled self-supervised learning, which can learn meaningful node representations on non-homophilous graphs by capturing the intrinsic graph structure. The core idea of DSSL is to model the distributions of node neighbors via a mixture of generative processes in the representation space. Specifically, we model the generation of neighbors by assuming each node has latent heterogeneous factors which are utilized to make connections to its different neighbors. Intuitively, the factor denotes various reasons behind why two nodes are connected in non-homophilous graphs. For instance, two nodes in a school network will be connected depending on some factors such as colleagues, friends, or classmates; in protein networks, even if they do not have similar features, different amino acid types are likely to be connected due to various interactions.

Formally, let $f_\theta(\mathbf{X}, \mathbf{A})[i] = \mathbf{v}_i$ and $f_\theta(\mathbf{X}, \mathbf{A})[j] = \mathbf{z}_j$ be the representations of nodes $v_i$ and $v_j$, respectively. Here we utilize different notations, i.e., $\mathbf{v}$ and $\mathbf{z}$ to distinguish between central and neighbor nodes since each node plays two roles: the node itself and a specific neighbor of other nodes. Our goal is to find the encoder parameter $\theta$ which maximizes the likelihood of distribution $p(\mathbf{z}_j|\mathbf{v}_i; \theta)$ on central node and its neighbor. To model the unobserved factors, we associate every node $\mathbf{v}_i$ with a discrete latent variable $k$ to indicate to which factor $\mathbf{v}_i$ has. Assume that there are $K$ factors in total, the log-likelihood of node neighbors $\mathbf{v}_i$ can be written by marginalizing out the latent variables:

$$\mathcal{L}_{DSSL}(\theta) = \frac{1}{|N(i)|} \sum_{j \in N(i)} \log[p_\theta(\mathbf{z}_j|\mathbf{v}_i)] = \frac{1}{|N(i)|} \sum_{j \in N(i)} \log\Big[\sum_{k=1}^{K} p_\theta(\mathbf{z}_j|\mathbf{v}_i, k)p_\theta(k|\mathbf{v}_i)\Big], \quad (1)$$

where $N(i)$ is the set of out-neighbors of $v_i$, the distribution $p_\theta(k|\mathbf{v}_i)$ indicates the assignment of latent semantics over central node representation $\mathbf{v}_i$, and $p_\theta(\mathbf{z}_j|\mathbf{v}_i, k)$ is the probability that node $v_i$ and its neighbor $v_j$ are connected under factor $k$. Unlike previous works [46, 14, 26], which directly encourage nearby nodes to have similar representations, we provide an alternative way to model node neighbors and seek to decouple their latent relationship without any prior on neighbor partitions. The similar high-level ideas of modeling latent variables between two nodes has also been explored in local augmentation in graphs (LA-GCN) [31] and vGraph [43]. Different from them, we consider modeling discrete relations in the learned representation space for self-supervised learning on non-homophilous graphs. In Eq. (1), probabilities $p_\theta(k|\mathbf{v}_i)$ and $p_\theta(\mathbf{z}_j|\mathbf{v}_i, k)$ are not specified, and involve discrete latent variables. To make it solvable, we introduce the following generative process.

Let $\boldsymbol{\mu}_k$ and $\boldsymbol{\Sigma}_k$ be the mean and variance of the latent mixture component $k$, and $\pi_k$ be its corresponding mixture probability. The generation of a node and its neighbor shown in Figure 2 (a) typically involves three steps: (1) draw a latent variable $k$ from a categorical distribution $p(k)$ on all mixture components, where $p(k)$ is usually defined as uniform distribution $p(k) = \frac{1}{K}$ for unknown graphs and better generalization, (2) draw the central node representation $\mathbf{v}_i$ from the Gaussian distribution $p_\theta(\mathbf{v}_i|k) = \mathcal{N}(\mathbf{v}_i; \boldsymbol{\mu}_k, \boldsymbol{\Sigma}_k)$, and (3) draw the neighbor representation $\mathbf{z}_j$ from Gaussian distribution $p_\theta(\mathbf{z}_j|\mathbf{v}_i, k) = \mathcal{N}(\mathbf{z}_j; \boldsymbol{\mu}_{z_j}, \boldsymbol{\Sigma}_j)$ where the mean depends both on central representation and latent variable: $\boldsymbol{\mu}_{z_j} = \mathbf{v}_i + \beta g_\theta(k)$. Here the projector $g_\theta(\cdot)$ denotes another network that embeds latent variable $k$ to the representation space, and $\beta$ is the parameter that controls the strength of interpolation. In practice, to reduce complexity, we consider using isotropic Gaussian, i.e., $\forall k: \Sigma_k = \mathbf{I}\sigma_1^2$ and $\forall j: \Sigma_j = \mathbf{I}\sigma_2^2$ where $\mathbf{I}$ is the identity matrix, $\sigma_1$ and $\sigma_2$ are hyperparameters. In alignment with this generative process, the joint distribution of the edge and latent variable can be written as:

$$p_\theta(\mathbf{v}_i, \mathbf{z}_j, k) = p_\theta(\mathbf{z}_j|\mathbf{v}_i, k)p_\theta(\mathbf{v}_i|k)p(k). \quad (2)$$

Intuitively, $p_\theta(\mathbf{v}_i|k)$ can be viewed as the probability for node representation under the $k$-th mixture component. Regarding $p_\theta(\mathbf{z}_j|\mathbf{v}_i, k) = \mathcal{N}(\mathbf{z}_j; \boldsymbol{\mu}_{z_j}, \boldsymbol{\Sigma}_j)$, this formulation is inspired by knowledge graph embedding [3], where two entities should be closed to each other under a certain relation operation. However, unlike knowledge graph embedding, the relational context is a latent variable and not observed in our setting. Intuitively, instead of enforcing the exact representation alignment of two linked nodes [26, 14, 17, 16, 46], our design can relax this strong homophily assumption and account for the semantic shift of representations between the central node and its neighbors.

Now, the posterior probability $p_\theta(k|\mathbf{v}_i)$ in Eq. (1) can be derived by using Bayesian rules as follows:

$$p_\theta(k|\mathbf{v}_i) = \frac{p_\theta(\mathbf{v}_i|k)p(k)}{\sum_{k'=1}^{K} p_\theta(\mathbf{v}_i|k')p(k')} = \frac{\mathcal{N}(\mathbf{v}_i; \boldsymbol{\mu}_k, \mathbf{I}\sigma_1^2)p(k)}{\sum_{k'=1}^{K} \mathcal{N}(\mathbf{v}_i; \boldsymbol{\mu}_{k'}, \mathbf{I}\sigma_1^2)p(k')}, \quad (3)$$

where $\boldsymbol{\mu} = \{\boldsymbol{\mu}_k\}_{k=1}^K$ can be treated as a set of trainable prototype representations, which are the additional distribution parameters. This posterior probability represents the soft semantic assignment of the learned representation to the prototypes, which makes the node with similar semantic properties be close to its prototype and encode both the inter- and intra-cluster variation. Substituting $p_\theta(k|\mathbf{v}_i)$ and $p_\theta(\mathbf{z}_j|\mathbf{v}_i, k)$ in Eq. (1) with the specified probabilities, we get final objective $\mathcal{L}_{DSSL}(\theta, \boldsymbol{\mu})$.

### 3.3 Evidence Lower Bound

Given the framework above, we are interested in: (i) learning the model parameters $\theta$ and $\boldsymbol{\mu}$ by maximizing the log-likelihood $\mathcal{L}_{DSSL}(\theta, \boldsymbol{\mu})$, and (ii) inferring the posterior of latent variable $k$ for each observed links. However, it is computationally intractable to directly solve these two problems due to the latent variables. To solve this, we resort to amortized variational inference methods [25], and maximize the evidence lower-bound (ELBO) of Eq. (1), i.e., $\mathcal{L}_{\text{DSSL}}(\theta, \boldsymbol{\mu}) \geq \mathcal{L}_{\text{DSSL}}(\theta, \phi, \boldsymbol{\mu})$:

$$\mathcal{L}_{\text{DSSL}}(\theta, \phi, \boldsymbol{\mu}) = \frac{1}{|N(i)|} \sum_{j \in N(i)} \mathbb{E}_{q_\phi(k|\mathbf{v}_i, \mathbf{z}_j)}[\log p_\theta(\mathbf{z}_j|\mathbf{v}_i, k) + \log p_\theta(k|\mathbf{v}_i)] + \mathcal{H}(q_\phi(k|\mathbf{v}_i, \mathbf{z}_j)), \quad (4)$$

where $q_\phi(k|\mathbf{v}_i, \mathbf{z}_j)$ is the introduced variational distribution parameterized by $\phi$ and $\mathcal{H}(\cdot)$ is the entropy operator. $\mathbf{z}_j$ is included in this variational posterior, so the inference is also conditioned on the neighborhood information. We derive this ELBO in Appendix A.1. Maximizing this ELBO w.r.t. $\{\theta, \phi, \boldsymbol{\mu}\}$ is equivalent to (i) maximizing $\mathcal{L}_{\text{DSSL}}(\theta, \boldsymbol{\mu})$ and to (ii) make variational $q_\phi(k|\mathbf{v}_i, \mathbf{z}_j)$ be close to true posterior. Plugging the parameterized probabilities into this ELBO, we obtain the following loss to minimize. See Appendix A.2 for corresponding derivations.

$$\mathcal{L} = \frac{1}{|N(i)|} \sum_{j \in N(i)} \mathbb{E}_{q_\phi(k|\mathbf{v}_i, \mathbf{z}_j)}\left[\|\mathbf{v}_i + \beta g_\theta(k) - \mathbf{z}_j\|_2^2\right] - \sigma_2^2 \mathbb{E}_{q_\phi(k|\mathbf{v}_i)}\left[\log \frac{\exp(\mathbf{v}_i^\top \cdot \boldsymbol{\mu}_k / \sigma_1^2)}{\sum_{k'=1}^K \exp(\mathbf{v}_i^\top \cdot \boldsymbol{\mu}_{k'} / \sigma_1^2)}\right]$$
$$- \mathcal{H}(q_\phi(k|\mathbf{v}_i, \mathbf{z}_j)), \quad (5)$$

where $q_\phi(k|\mathbf{v}_i) = 1/|N(i)| \sum_{j \in N(i)} q_\phi(k|\mathbf{v}_i, \mathbf{z}_j)$ is the posterior probability of semantic assignment for central node $v_i$, by aggregating all its neighbors. Thus, the first term (denoted as $\mathcal{L}_{local}$) in the loss encourages the model to reconstruct the local neighbors while considering different semantic shifts captured by latent variable $k$ (see Figure 2 (b)). The second term (denoted as $\mathcal{L}_{global}$) encourages the model to perform clustering with learned representation where possible, i.e., seeking to push the representation $\mathbf{v}_i$ to its closest prototype cluster (see Figure 2 (c)). The final entropy term makes the model choose to have high entropy over $q_\phi(k|\mathbf{v}_i, \mathbf{z}_j)$ such that all of the $K$-channel losses must be low. Overall, this loss derived from ELBO can capture global semantic similarities over neighborhoods and learn to decouple different latent patterns in the local neighbors.

Regarding the variaitonal distribution $q_\phi(k|\mathbf{v}_i, \mathbf{z}_j)$, we model it as categorical distribution since $k$ is a discrete multinomial variable. Specifically, the representations $\mathbf{v}_i$ and $\mathbf{z}_j$ are encoded to a combined representation and then $q_\phi(k|\mathbf{v}_i, \mathbf{z}_j)$ is determined by an output softmax inference head as follows:

$$q_\phi(k|\mathbf{v}_i, \mathbf{z}_j) = \frac{\exp(h_\phi([\mathbf{v}_i; \mathbf{z}_j])[k])}{\sum_{k'=1}^K \exp(h_\phi([\mathbf{v}_i; \mathbf{z}_j])[k'])}, \quad (6)$$

where $h_\phi$ denotes the inference network parameterized by $\phi$ and $[\cdot, \cdot]$ can be the element-wise product or concatenation operation. $h_\phi([\mathbf{v}_i; \mathbf{z}_j])[k]$ indicates the $k^{th}$ element, i.e., the logit corresponding the latent context $k$. Instead of introducing variational parameters individually, we consider the amortization inference [25, 58], which fits a shared network to calculate each local parameter.

For the expectation terms in Eq. (5), back-propagation through the discrete variable $k$ is not directly feasible. We alleviate this by adopting the Straight-Through Gumbel-Softmax estimator [21], which provides a continuous differentiable approximation for drawing samples from a categorical distribution. Specifically, for each sample, a latent cluster vector $\mathbf{c} \in (0, 1)^K$ is drawn from:

$$\mathbf{c}[k] = \frac{\exp((h_\phi([\mathbf{v}_i; \mathbf{z}_j])[k] + \epsilon[k]) / \gamma)}{\sum_{k'=1}^K \exp((h_\phi([\mathbf{v}_i; \mathbf{z}_j])[k] + \epsilon[k']) / \gamma)}, \quad (7)$$

where $\epsilon[k]$ is i.i.d drawn from the $\text{Gumbel}(0, 1)$ distribution and $\gamma$ is a temperature. With this reparameterization trick, we can obtain the surrogate $\mathbb{E}_{q_\phi(k|\mathbf{v}_i, \mathbf{z}_j)}[\|\mathbf{v}_i + \beta g_\theta(k) - \mathbf{z}_j\|_2^2] \simeq \mathbb{E}_\epsilon[\|\mathbf{v}_i + \beta g_\theta(\mathbf{c}) - \mathbf{z}_j\|_2^2]$ and the gradients are estimated with Monte Carlo. The expectation term over $q_\phi(k|\mathbf{v}_i)$ can be similarly estimated. Then, $\{\theta, \phi, \boldsymbol{\mu}\}$ in Eq. (5) can be efficiently solved by gradient descent.

## 3.4 Algorithm Optimization

The overall optimization involves simultaneously training (1) the encoder $f_\theta$, (2) the projector $g_\theta$, (3) the inference predictor $h_\phi$ and (4) the prototypes $\boldsymbol{\mu}$. The most canonical way to update the parameters is stochastic gradient descent. However, we observe that stochastically updating all parameters suffers from two problems: (1) The objective admits trivial solutions, e.g., outputting the same representation for all nodes in the optimization process. (2) Updating prototypes without any constraints will lead to a degenerate solution, i.e., all nodes are assigned to a single cluster.

To address the issue of trivial solutions, inspired by the recent works [19, 13], we consider an asymmetric encoder architecture that includes online and target encoders. Specifically, for each node pair $(v_i, v_j)$, the online encoder $f_\theta$ produces the representation of the central node $\mathbf{v}_i$; while the target encoder $f_\xi$ is used to produce the representation of its neighbor $\mathbf{z}_j$. Importantly, the gradient of loss is only used to update the online encoder $f_\theta$ while being blocked in the target encoder. The weights of the target encoder $\xi$ are updated via the exponential moving average (EMA) of the online encoder $\theta$:

$$[\theta, \phi, \boldsymbol{\mu}] \leftarrow [\theta, \phi, \boldsymbol{\mu}] - \boldsymbol{\Gamma}\left(\nabla_{\theta,\phi,\boldsymbol{\mu}}\mathcal{L}\right), \quad \xi \leftarrow \tau\xi + (1-\tau)\theta, \tag{8}$$

where $\boldsymbol{\Gamma}(\cdot)$ indicates a stochastic optimizer and $\tau \in [0,1]$ is the target decay rate. This update introduces an asymmetry between two encoders that prevents collapse to trivial solutions [13, 48]. To alleviate the second issue, besides the stochastic update, we also apply a global update for the prototype vectors $\boldsymbol{\mu} = \{\boldsymbol{\mu}_i\}_{i=1}^K$ at the end of each training epoch to avoid a degenerate solution:

$$\boldsymbol{\mu}_k = \frac{\sum_{i=1}^N \pi_i(k) \cdot \mathbf{v}_i}{\|\sum_{i=1}^N \pi_i(k) \cdot \mathbf{v}_i\|_2^2}, \quad \text{where} \ \ \pi_i(k) = 1/|N(i)| \sum_{j \in N(i)} q_\phi(k|\mathbf{v}_i, \mathbf{z}_j). \tag{9}$$

The derivation is provided in Appendix A.3. Intuitively, $\pi_i(k)$ reflects the degree of relevance of node $v_i$ to the $k^{th}$ prototype. Instead of only updating the prototypes in a mini-batch, we also aggregate all the representations as the prototype based on the soft assignment probability at the end of the epoch. After the training is finished, we only keep the online encoder $f_\theta$ for the downstream task. Our full algorithm and network are provided in Appendix B. The details about time complexity of DSSL is given in Appendix C.1, which scales linearly in the size of edges.

## 3.5 Theoretical Analysis

In this section, we provide an analysis of the proposed framework. We first present the connection between the proposed objective and the mutual information maximization, then show that the learned representations by our objective provably enjoy a good downstream performance. Due to the space limitation, all proofs of theorems and corollaries are provided in Appendix D.

We denote the random variable of the input graph as $\mathcal{G}$ and the downstream label as $\mathbf{y}$. For clarity, we omit subscript $i$ in what follows. From an information-theoretic learning perspective, a desirable way is to maximize the mutual information $I(\mathbf{v}, \mathbf{y})$ between the representation $\mathbf{v}$ and downstream label $\mathbf{y}$. However, due to the lack of the downstream label, self-supervised learning resorts to maximizing $I(\mathbf{v}, \mathbf{s})$ where $\mathbf{s}$ is different designed self-supervised signal [51, 4, 56, 65, 11]. In our method, we have two self-supervised signals: the global semantic cluster information inferred by $k$ and the local structural roles captured by the representations of the neighbors $\mathbf{z} = \{\mathbf{z}_i | v_i \in N(v)\}$ of node $v$.

Then, we can interpret our objective in Eq. (5) from the information maximization perspective:

**Theorem 1.** *Optimizing local and global terms in Eq. (5) is equivalent to maximizing the mutual information between the representation $\mathbf{v}$ and global signal $k$ and maximizing the conditional mutual information between $\mathbf{v}$ and the local signal $\mathbf{z}$, conditioned on global signal $k$. Formally, we have:*

$$\min_{\theta,\phi,\boldsymbol{\mu}} \mathcal{L} \Rightarrow \max_{\mathbf{v}} I(\mathbf{v}; k) + I(\mathbf{v}; \mathbf{z}|k) = I(\mathbf{v}; k, \mathbf{z}). \tag{10}$$

This theorem suggests that we essentially combine both local structure and global semantic information as the self-supervised signal and maximize the mutual information between the representation $\mathbf{v}$ and their joint distribution $(k, \mathbf{z})$. Next, we discuss how the learned representation affects the downstream task $\mathbf{y}$ based on the information bottleneck principle [51, 11, 56, 4]. The rationality of self-supervised learning is that the task-relevant information lies mostly in the shared information between the input and the self-supervised signals [51, 11, 4]. Specifically, we formulate our lightweight and reasonable assumption below, which serves as a foundation for our analysis.

**Assumption 1.** *Nodes with similar labels should have similar "local structural roles" and "global semantic clusters." In this work, we equate "local structure" with the 1-hop neighborhood and "global*

*semantic" with the clustering membership of a node. Formally, we have the task-relevant information* $\mathbf{y}$ *left in* $\mathcal{G}$ *except that in joint self-supervised signal* $(\mathbf{z}, k)$ *is relatively small:* $I(\mathcal{G}; \mathbf{y}|\mathbf{z}, k) \leq \epsilon$.

Intuitively, this assumption indicates that most of the task-relevant information in the graph is contained in the self-supervised signal (local neighborhood and global semantic patterns). Based on this assumption, we have the following theorem, which reveals why the downstream tasks can benefit from the learned representations learned by our objective function.

**Theorem 2.** *Let* $\mathbf{v}_{\text{joint}} = \arg\max_{\mathbf{v}} I(\mathbf{v}; \mathbf{z}, k), \mathbf{v}_{\text{local}} = \arg\max_{\mathbf{v}} I(\mathbf{v}; k),$ *and* $\mathbf{v}_{\text{global}} = \arg\max_{\mathbf{v}} I(\mathbf{v}; \mathbf{z})$. *Formally, we have the following inequalities about the task-relevant information:*

$$I(\mathcal{G}; \mathbf{y}) = \max_{\mathbf{v}} I(\mathbf{v}; \mathbf{y}) \geq I(\mathbf{v}_{\text{joint}}; \mathbf{y}) \geq \max(I(\mathbf{v}_{\text{local}}; \mathbf{y}), I(\mathbf{v}_{\text{global}}; \mathbf{y})) \geq I(\mathcal{G}; \mathbf{y}) - \epsilon. \quad (11)$$

Theorem 2 shows that the gap of task-relevant information between supervised representation $\mathbf{v}_{sup} = \arg\max_{\mathbf{v}} I(\mathbf{v}, \mathbf{y})$ and self-supervised representation $\mathbf{v}_{\text{joint}}$ is $\epsilon$. Thus, we can guarantee a good downstream performance as long as the Assumption 1 is satisfied. It is noteworthy that jointly utilizing local structure and global semantics as the self-supervised signal is expected to contain more task information. As further enlightenment, we can relate Eq. (36) with the Bayes error rate [51]:

**Corollary 1.** *Suppose that downstream label* $\mathbf{y}$ *is a M-categorical random variable. Then we have the upper bound for the downstream Bayes errors* $P_{\mathbf{v}}^e = \mathbb{E}_{\mathbf{v}} \left[ 1 - \max_{y \in \mathbf{y}} P(\hat{y} = y|\mathbf{v}) \right]$ *on learned representation* $\mathbf{v}$*, where* $\hat{\mathbf{y}}$ *is the estimation for label from our downstream classifier:*

$$\text{Th}(P_{\mathbf{v}_{joint}}^e) \leq \log 2 + P_{\mathbf{v}_{\text{sup}}}^e \cdot \log M + I(\mathcal{G}; \mathbf{y}|\mathbf{z}, k) \triangleq \text{RHS}_{\mathbf{v}_{joint}}, \quad (12)$$

*where* $\text{Th}(x) = \min\{\max\{x, 0\}, 1 - 1/|M|\}$ *is a thresholded operation [51]. Similarly, we can obtain the error upper bound of other representations* $\mathbf{v}_{\text{local}}$ *and* $\mathbf{v}_{\text{global}}$ *:* $\text{RHS}_{\mathbf{v}_{local}}$ *and* $\text{RHS}_{\mathbf{v}_{global}}$*. Then, we have inequalities on error upper bounds* $: \text{RHS}_{\mathbf{v}_{joint}} \leq \min(\text{RHS}_{\mathbf{v}_{local}}, \text{RHS}_{\mathbf{v}_{global}})$*.*

This corollary says that our self-supervised signal has a tighter upper bound on the downstream Bayes error. Thus, we can expect that the representation learned by our objective function, which utilizes both local structure and global semantic information, has superior performance on downstream tasks.

## 4 Experiments

In this section, we empirically evaluate the proposed self-supervised learning method on several real-world graph datasets and analyze its behavior on graphs to gain further insights.

### 4.1 Experimental Setup

**Datasets.** We perform experiments on widely-used homophilic graph datasets: Cora, Citeseer, and Pubmed [42], as well as non-homophilic datasets: Texas, Cornell, Wisconsin [37], Penn94 and Twitch. Penn94 and Twitch are two relatively large non-homophilous graph datasets proposed by [41, 29]. We provide the detailed descriptions, statistics, and homophily measures of datasets in Appendix E.1.

**Baselines.** To evaluate the effectiveness of DSSL, we consider the following representative unsupervised and self-supervised learning methods for the node representation task, including Deepwalk [38], LINE [46], Struc2vec [40], GAE [26], VGAE [26], DGI [53], GraphCL [64], MVGRL [18] and BGRL [48]. The detailed description of baselines and implementations are given in Appendix E.2.

**Evaluation Protocol.** We consider two types of downstream tasks: node classification and node clustering. For node classification, we follow the standard linear-evaluation protocol on graphs [53, 48], where a linear classifier is trained on top of the frozen representation, and test accuracy (ACC) is used as a proxy for representation quality. For datasets, we adopt the similar random split with a train/validation/test split ratio of 48/32/20% for the training of downstream linear classifier following [37]. Note that [37] claims that the ratios are 60/20/20%, which is different from the actual shared data splits as shown by [68, 34, 29]. For the node clustering task, we perform $K$-means clustering on the obtained representations and set the number of clusters to the number of classes. We utilize the normalized mutual information (NMI) [54] as the evaluation metric for clustering.

**Setup.** For all self-supervised methods on graph neural networks, we consider a two-layer GCN [27] as the encoder unless otherwise stated, and randomly initialize parameters. We run experiments with 10 random splits and report the average performance. We select the best configuration of hyperparameters based on accuracy on the validation. The detailed settings are given in Appendix E.2.

Table 1: Experimental results (%) with standard deviations on the node classification task. The best and second best performance under each dataset are marked with boldface and underline, respectively.

| Method | Cora | Citeseer | Pubmed | Texas | Cornell | Squirrel | Penn94 | Twitch |
|---|---|---|---|---|---|---|---|---|
| Deepwalk | $77.14_{\pm0.82}$ | $67.85_{\pm0.79}$ | $79.38_{\pm1.22}$ | $42.31_{\pm2.21}$ | $41.55_{\pm3.12}$ | $37.54_{\pm2.19}$ | $56.13_{\pm0.46}$ | $66.37_{\pm0.11}$ |
| LINE | $78.93_{\pm0.55}$ | $68.79_{\pm0.41}$ | $80.56_{\pm0.92}$ | $48.69_{\pm1.39}$ | $43.68_{\pm2.17}$ | $38.92_{\pm1.58}$ | $57.59_{\pm0.17}$ | $67.23_{\pm0.27}$ |
| Struc2vec | $30.26_{\pm1.52}$ | $53.38_{\pm0.62}$ | $40.83_{\pm1.85}$ | $49.31_{\pm3.22}$ | $30.22_{\pm5.87}$ | $36.49_{\pm1.15}$ | $50.29_{\pm0.31}$ | $63.52_{\pm0.35}$ |
| GAE | $78.33_{\pm0.27}$ | $66.39_{\pm0.24}$ | $78.28_{\pm0.77}$ | $53.98_{\pm3.22}$ | $44.18_{\pm3.56}$ | $30.53_{\pm1.33}$ | $58.11_{\pm0.18}$ | $67.98_{\pm0.27}$ |
| VGAE | $80.59_{\pm0.35}$ | $69.90_{\pm0.57}$ | $81.33_{\pm0.69}$ | $50.27_{\pm2.21}$ | $48.73_{\pm4.19}$ | $29.13_{\pm1.16}$ | $58.29_{\pm0.21}$ | $65.09_{\pm0.08}$ |
| DGI | $84.17_{\pm1.35}$ | $71.80_{\pm1.33}$ | $81.65_{\pm0.71}$ | $\underline{58.53}_{\pm2.98}$ | $45.33_{\pm6.11}$ | $26.44_{\pm1.12}$ | $53.68_{\pm0.19}$ | $66.97_{\pm0.25}$ |
| GraphCL | $84.28_{\pm0.91}$ | $72.46_{\pm1.79}$ | $81.96_{\pm0.73}$ | $48.67_{\pm4.37}$ | $47.22_{\pm4.50}$ | $22.53_{\pm0.98}$ | $58.43_{\pm0.31}$ | $68.37_{\pm0.16}$ |
| MVGRL | $\mathbf{85.21}_{\pm1.18}$ | $72.13_{\pm1.04}$ | $82.33_{\pm0.88}$ | $51.26_{\pm0.38}$ | $\underline{51.16}_{\pm1.67}$ | $38.43_{\pm0.87}$ | $57.22_{\pm0.17}$ | $66.03_{\pm0.26}$ |
| BGRL | $83.29_{\pm0.72}$ | $71.56_{\pm0.87}$ | $81.34_{\pm0.50}$ | $52.77_{\pm1.98}$ | $50.33_{\pm2.29}$ | $36.22_{\pm1.97}$ | $\underline{58.98}_{\pm0.13}$ | $67.43_{\pm0.22}$ |
| DSSL | $83.06_{\pm0.53}$ | $\mathbf{73.51}_{\pm0.64}$ | $\mathbf{82.98}_{\pm0.49}$ | $\mathbf{62.11}_{\pm1.53}$ | $\mathbf{53.15}_{\pm1.28}$ | $\mathbf{40.51}_{\pm0.38}$ | $\mathbf{60.38}_{\pm0.32}$ | $\mathbf{69.81}_{\pm0.17}$ |

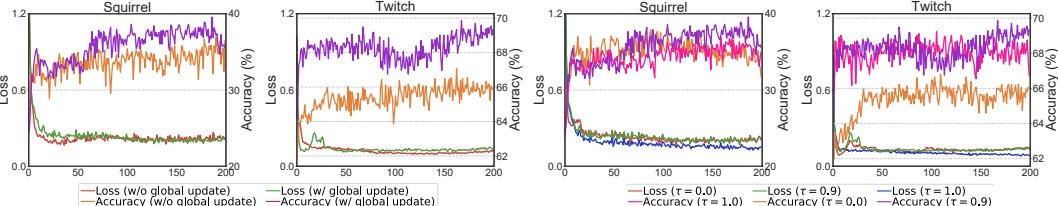

Figure 3: (Left) Performance w/ and w/o global update. (Right) Performance with varying $\tau$.

## 4.2 Overall Performance Comparison

In this section, we conduct experiments on real-world graphs compared to state-of-the-art methods. Table 1 reports the average classification accuracy with the standard deviation on node classification after ten runs. Since node clustering results have a similar tendency, we provide them in Appendix F.1. From Table 1, we have the following observations: (1) Generally, our DSSL achieves the best performance on both node clustering and classification tasks over the best baseline, demonstrating the effectiveness of DSSL on both homophilous and non-homophilous graphs and the robustness to different downstream tasks and graphs. Especially, DSSL achieves a relative improvement of over 6% and 4% on Texas and Squirrel compared to the best baselines (2) Our DSSL cannot achieve the best performance on Cora. This is reasonable since Cora is highly homophilous (see Appendix E.1), and the core design of augmentation in contrastive learning methods such as MVGRL and GraphCL enables them to be effective on homophilous graphs but failed on low-homophily settings. This supports our motivation in the introduction that it is relatively difficult to design effective graph augmentation on non-homophilous graphs due to their heterogeneous and diverse patterns. (3) When compared with GAE, VGAE, and DGI, DSSL consistently and significantly outperforms them. We deem that this improvement is mainly from the joint local semantic shift and global semantic clustering in DSSL, which is not included in existing methods.

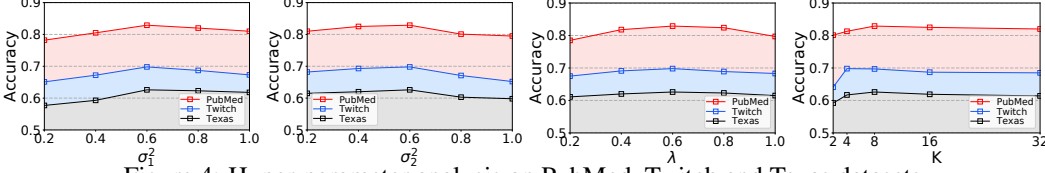

Figure 4: Hyper-parameter analysis on PubMed, Twitch and Texas datasets.

## 4.3 Ablation Study and Parameter Analysis

In this section, we conduct an ablation study and investigate the sensitivity of hyper-parameters.

**Ablation Study.** We consider the following ablations: **(A1)** We remove the key component of DSSL: the local reconstruction loss (`w/o` $\mathcal{L}_{local}$). **(A2)** We remove global clustering loss to see if $\mathcal{L}_{local}$ alone is still effective (`w/o` $\mathcal{L}_{global}$). **(A3)** We remove the entropy term in Eq. (5) (`w/o entropy`). **(A4)** We remove the semantic shift term $g_\theta(k)$ in $\mathcal{L}_{local}$ (`w/o semantic shift`). **(A5)** We set posterior $q_\phi(k|\mathbf{v}_i, \mathbf{z}_j) = 1/K$ as uniform distribution for each link (`w/ uniform posterior`). We show the ablation study results in Table 2. $\mathcal{L}_{global}$ alone does not provide much discriminative information, and it does not perform very well. $\mathcal{L}_{local}$ as a key component of DSSL alone produces better results

Table 3: Experimental results (%) on the node classification task with GAT. The best and second best performances under each dataset are marked with boldface and underline, respectively.

| Method | Cora | Citeseer | Pubmed | Texas | Cornell | Squirrel | Penn94 | Twitch |
|--------|------|----------|--------|-------|---------|----------|--------|--------|
| MVGRL | **84.52**$_{\pm0.95}$ | 71.52$_{\pm0.41}$ | 81.05$_{\pm0.68}$ | 53.42$_{\pm0.29}$ | 50.29$_{\pm0.96}$ | 37.58$_{\pm0.95}$ | 57.21$_{\pm0.30}$ | 67.11$_{\pm0.23}$ |
| BGRL | 83.91$_{\pm0.25}$ | 72.15$_{\pm0.42}$ | 80.12$_{\pm0.52}$ | 51.02$_{\pm1.10}$ | 48.97$_{\pm1.03}$ | 35.33$_{\pm1.12}$ | 56.52$_{\pm0.25}$ | 66.21$_{\pm0.33}$ |
| DSSL | 82.54$_{\pm0.46}$ | **73.67**$_{\pm0.68}$ | **81.69**$_{\pm0.37}$ | **59.73**$_{\pm1.28}$ | **53.63**$_{\pm1.16}$ | **39.42**$_{\pm1.34}$ | **58.97**$_{\pm0.26}$ | **69.55**$_{\pm0.48}$ |

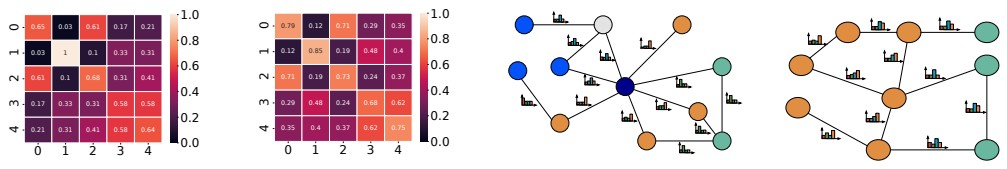

(a) Neighborhood similarity on Texas  (b) Latent factor similarity on Texas  (c) Visualization of subgraph (Texas id 58)  (d) Visualization of subgraph (Twitch id 9)

Figure 5: The visualization and case study results (best viewed on a computer screen and note that the latent distribution of each link need to be zoomed in to be better visible).

than the DGI baseline. We can also observe that considering semantic shift and personalized posterior can further improve the performance a lot which demonstrates our key motivations. The full model (last row) achieves the best performance, which illustrates that different components in the proposed DSSL are complementary to each other.

**Effect of Global Update on Prototypes**. We conduct ablation studies to gain insights into the global updating of prototypes. As shown in Figure 3, we can observe that without Eq. (9), the performance is not very good, and we are stuck in bad local optima. As discussed above, a possible reason is that we will experience strong degeneracy if we only update prototype vectors with mini-batch training. This figure also demonstrates that DSSL can converge within a few hundred steps, which is efficient.

**Effect of Asymmetric Architecture**. We then explore the effect of target decay rate $\tau$ on the performance. Figure 3 shows the learning curves of DSSL on Squirrel and Twitch. We can observe that the best result is achieved at $\tau = 0.9$. When $\tau = 1.0$, i.e., the target network is never updated, DSSL obtains a competitive result but is lower than $\tau = 0.9$. This confirms that slowly updating the target network is crucial in obtaining superior performance. At the other extreme value $\tau = 0$, the target network is the same as the online network, and DSSL demonstrates a degenerated performance.

Table 2: Node classification accuracy (%) on the Texas dataset.

| Ablation | Accuracy |
|----------|----------|
| **A1** w/o local loss $\mathcal{L}_{local}$ | 25.18$\pm$1.31 |
| **A2** w/o global loss $\mathcal{L}_{global}$ | 59.34$\pm$2.76 |
| **A3** w/o entropy loss | 60.57$\pm$1.27 |
| **A4** w/o semantic shift | 56.19$\pm$1.25 |
| **A5** w/ uniform posterior | 50.27$\pm$1.04 |
| **A2+A3** | 57.61$\pm$1.72 |
| **A2+A4** | 50.52$\pm$2.01 |
| **A2+A5** | 48.59$\pm$1.87 |
| DGI | 58.53$\pm$2.98 |
| DSSL | **62.11**$\pm$**1.53** |

**Hyper-parameters Analysis.** We investigate the hyper-parameters most essential to our framework design, i.e., the standard deviation $\sigma_1$ and $\sigma_2$, the temperature of the Gumbel Softmax $\gamma$, and the total number of factors $K$. The corresponding results are shown in Figure 4. $\sigma_1^2$ resembles the temperature scaling in Eq. (5). We observe $\sigma_1^2$ is better to be selected from 0.6 to 1.0, and a too small (e.g., 0.2) value may degenerate the performance in all three datasets. $\sigma_2^2$ balances the local and global loss in Eq. (5). We can find that having large values of $\sigma_2^2$ does not improve the performance, as the local loss plays an essential role in the proposed model. Further, we observe that DSSL is not very sensitive to the Gumbel softmax temperature $\lambda$, while a moderate hardness of the Softmax gives the best results. We also find that as $K$ increases from 2 to 8, the performance of DGCL improves, which suggests the importance of decoupling latent factors. However, training with large $K$ will lead to a performance slightly drop. We provide results on other datasets in Appendix F.3.

**Encoders Analysis.** To further evaluate the effectiveness of the proposed DSSL, we consider the case where the self-supervised learning methods are implemented using another GNN encoder. Table 3 shows the results of the selected baseline with GAT as the encoder. As shown in Table 3, our DSSL performs better than all the compared methods in most cases, which once again proves the effectiveness of DSSL, and also shows that DSSL is broadly applicable to various GNN encoders.

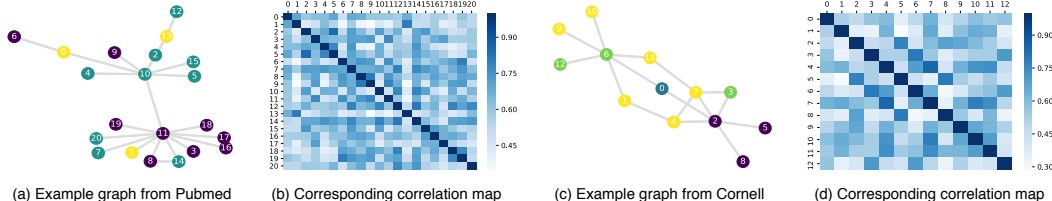

| (a) Example graph from Pubmed | (b) Corresponding correlation map | (c) Example graph from Cornell | (d) Corresponding correlation map |

Figure 6: Example graphs and correlation maps. The graphs are randomly sampled from Pubmed and Cornell. The correlation maps are obtained based on the pair-wise similarities of learned posteriors.

## 4.4 Visualization and Case Study

**Local neighborhood Patterns**. We provide the visualization and case study results to understand how DSSL uncovers the latent patterns in local neighborhoods. More results can be found in Appendix F.4. In Figure 5 (a), we calculate the cross-class neighborhood similarity (see Appendix E.3 for definition), which serves as the ground truth. If nodes with the same label share the same neighborhood distributions, the intra-class similarity should be high. In Figure 5 (b), we calculate average posterior distribution $q_\phi(k|\mathbf{v}_i)$ over all nodes on each class and provide cosine similarity of them. We can observe that our learned distribution shares a similar pattern with the cross-class neighborhood similarity, which shows that learned latent factors can capture the latent semantic information related to neighborhood patterns. In Figures 5 (c) and (d), we plot the subgraph of a randomly selected node, and the distribution $q_\phi(k|\mathbf{v}_i, \mathbf{z}_j)$. We use different colors to indicate different labels. We find that similar neighbors over class generally have a similar distribution, while different types of links exhibit different latent distributions. This observation matches our motivation that DSSL can decouple the diverse semantics in the local neighborhoods.

**Global Semantic Patterns.** We investigate how our learned semantic clusters perform on the long-range nodes. To show this, we study the learned cluster distribution $q_\phi(k|\mathbf{v}_i)$ for each node. To facilitate visualization, we randomly sample a sub-graph that contains high-hop neighborhoods from each dataset and use different colors to indicate different node labels. We then calculate the pair-wise similarity of the posteriors $q_\phi(k|\mathbf{v}_i)$ between nodes as shown in Figure 6. We can observe that our learned semantic clusters exhibit similar patterns for the nodes in the same class. Some nodes exhibit similar semantic clusters regardless of the distance between them. For instance, node ID 6 exhibits a very similar posterior distribution to the same class node ID 19 in Pubmed, and node ID 3 exhibits similar posterior distribution to node ID 12 in Cornell, despite these nodes being ID 4 hops away. Since our DSSL seeks to pull the node representation to the inferred semantic clusters, we can learn the global node-node relationships in representations instead of favoring nearby nodes.

## 5 Conclusions

In this paper, we study the problem of conducting self-supervised learning on non-homophilous graphs data. We present a novel decoupled self-supervised learning (DSSL) framework to decouple the diverse neighborhood context of a node in an unsupervised manner. Specifically, DSSL imitates the generative process of neighbors and explicitly models unobserved factors by latent variables. We show that DSSL can simultaneously capture the global clustering information and the local structure roles with the semantic shift. We theoretically show that DSSL enjoys a good downstream performance. Extensive experiments on several graph datasets validated the superiority of DSSL and showed that DSSL could learn meaningful class-discriminative representations.

## Acknowledgments

This work was partially supported by the National Science Foundation (NSF) under grants number IIS-1909702 and IIS-1955851, Army Research Office (ARO) under grant number W911NF-21-1-0198 and Department of Homeland Security under grant number E205949D. The findings and conclusions in this paper do not necessarily reflect the view of the funding agency.

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
