# OpenReview forum: "Decoupled Self-supervised Learning for Graphs"
_NeurIPS.cc/2022/Conference — NeurIPS 2022 Accept_

### Official Review · Reviewer_7ey5 · 2022-07-06

**Rating:** 8
**Confidence:** 4
**Soundness:** 4 excellent
**Presentation:** 4 excellent
**Contribution:** 4 excellent

**Summary:**

This work studies the new problem of self-supervised learning for node representations on non-homophilous graphs, with the novel latent variable modeling to capture local and global semantics of graphs by decoupling latent factors, which is supported by theoretical and empirical results. In particular,
* The authors first point out that self-supervised learning on non-homophilous graphs is undiscovered, yet it should be differently handled unlike in the existing self-supervised learning methods for homophilous graphs, due to a unique structure of non-homophilous graphs that connected nodes do not share similar properties.
* To tackle this problem, the authors model the latent variable based on the motivation that nodes with similar neighborhood distributions should have similar representations, on which the model can capture the local semantics and global cluster structures by decoupling the neighborhood context.
* The authors make the proposed latent variable model learnable in an end-to-end fashion with the famous amortized variational inference scheme.
* To alleviate the issue that all node representations are collapsed into one representation and are assigned to one cluster, the authors leverage two tricks: 1) use the asymmetric encoding scheme for node representations (i.e., updating central and neighboring nodes differently); 2) use the outputs of node representations with their mixture probabilities to obtain prototype vectors for node clustering.
* The authors theoretically show that their proposed method maximizes the mutual information between the node representation and the joint distribution of suggested local and global latent variables, which is further analyzed in terms of the generalization bound to downstream tasks.
* The authors empirically verify that the proposed DSSL largely outperforms existing baselines, especially on non-homophilous graphs, and several analyses support their model designs as well as their claims.

**Questions:**

### Major Questions and Suggestions
* Regarding the first weakness in the above, the authors can address it by discussing the relevant work [1].
* Regarding the second weakness in the above, the authors can address it by showing results of capturing long-range dependencies, perhaps in a similar manner in Figure 5 of the case study but based on nodes far away from each other.

### Minor Questions and Suggestions
* Does the color in Figure 1 denote the class of nodes? It seems more explanations are needed for the illustration of Figure 1.
* The meaning of encoding intra-cluster variation in Line 172 is unclear. How does the proposed formulation in equation (3) capture the intra-cluster variation?
* A lack of explanation on the projector g_\theta in Line 209. It is better to explain the projector, for readers who are not familiar with self-supervised learning.
* The contents on page 14 should be removed.

### Typos
* In Line 293, we.
* In Line 346, case.
* In Line 361, (b) and (c) should be (c) and (d).


**Limitations:**

The authors provide the limitations and potential negative societal impact of their work in Section F of the supplementary file. I am satisfied with them.

**Strengths And Weaknesses:**

### Strengths
* This work discovers a new task of self-supervised learning for nodes on non-homophilous graphs, with enough explanations on its inherent challenges.
* This work is technically and mathematically sound. Specifically, this work tackles the proposed task with the novel and interesting latent variable formulation that models distributions of node's neighbors with decoupled latent factors, which can capture local distributions with reconstruction loss but also global cluster semantics with cluster centroids. Also, the authors theoretically show the effectiveness of the proposed DSSL in the view of information maximization.
* The empirical results of the proposed DSSL are significant, compared against relevant baselines.
* The authors comprehensively conduct the ablation study and sensitivity analysis.
* This paper is extremely well written and easy to follow.

### Weaknesses
Note that the below points are not the major weaknesses of this paper, but it would improve the paper if the authors address them adequately.
* There is a relevant work that, in graph self-supervised learning, [1] considers both local and global semantic structures, where the global context is captured by class prototypes which could be considered similar to the authors' method in the motivational perspective.
* The authors claim in the introduction section that, one challenge of graph self-supervised learning is to capture the long-range semantic dependency described in Line 66. However, I cannot find empirical results that the proposed DSSL framework can alleviate this issue. The case study in Figure 5 only shows the local dependencies between nearby nodes, not the long-range dependencies.

---

[1] Xu et al. Self-supervised Graph-level Representation Learning with Local and Global Structure. ICML 2021.

---

> ### Author Response · Authors · 2022-08-02
> **Response to Reviewer 7ey5**
>
> Thank you for the great summarization of our contributions, and we really appreciate your encouraging comments. Please see our responses below:
>
> **Q1. Discussion with the relevant work [1].**
>
> We thank the reviewer for  sharing this great work [1]. Here, we would like to discuss [1]  and clarify some important differences between [1] and our work.
>
> **Motivation**. Their goal is to conduct a graph-level classification task where each instance is a whole graph, and a dataset can be seen as a set of independent and identically distributed (IID) pairs. Thus the local structure in [1] means that, for a pair of similar (augmented) graphs, their embeddings are expected to be nearby in the latent space. In contrast, we focus on the node-level representation problem on non-homophilous graphs, which involves the interconnection among nodes and results in the non-IID nature of nodes. Thus, in our work, the "local" means each node's local neighbor distribution, and we propose to decouple the unobserved pattern in the local neighbors.
>
> **Framework**. The work [1] is mainly based on contrastive learning with noise-contrastive estimation (NCE), which requires prefabricated augmentations and negative samples. By contrast, our framework is a fully generative model and does not rely on any augmentations, and we decouple the diverse neighborhood context of a node with a generative process to capture the diverse local information. This makes our learning framework different from [1].
>
> **Theory**. We analyze the properties of DSSL and theoretically show why combining both local structure and global semantic information in the learned representations can achieve better downstream performance, which has not been mentioned and proposed by work [1].
>
> Following your suggestion, we have included the above discussion in Appendix G to give readers a more in-depth understanding of our work and [1].
>
> **Q2. Results of capturing long-range dependencies**.
>
> Thanks for your good suggestions, which are helpful for improving our paper. We have provided the cause studies to how our learned semantic clusters perform on the long-range nodes in Appendix F.7. Please refer to our revision for the update. From the results, we can find that some nodes exhibit similar semantic clusters learned by our DSSL, regardless of the distance between them.
>
> **Q3. Does the color in Figure 1 denote the class of nodes? It seems more explanations are needed for the illustration of Figure 1.**
>
> Yes! The color in Figure 1 denoted the classes of nodes. We apologize for the confusion and have provided some explanations in Figure 1.
>
>
> **Q4. The meaning of encoding intra-cluster variation in Line 172 is unclear. How does the proposed formulation in equation (3) capture the intra-cluster variation?**
>
> Thanks for your insightful question. The proposed formulation in equation (3) results in the global loss, i.e., the second term in equation (5). Specifically, the numerator in global loss seeks to pull the representation $v_{i}$ to the inferred cluster $\mu_{k}$ which optimizes the tightness of the clusters in the representation space and reduces the intra-cluster variation. Meanwhile, the denominator pushes all clusters away from $v_{i}$, which should cause them to be roughly uniformly distributed and increase the inter-cluster variation. We apologize for this confusion and will clearly state this in the main text of our final version.
>
> **Q5. A lack of explanation on the projector $g_\theta$ in Line 209?**
>
> Thanks for your question. In this work, the projector $g_\theta$ denotes a single fully connected network that embeds latent variable k (one-hot vector) to the node representation space. Please see line 156 and Figure 6.
>
> **Q6. Typos and contents on page 14 should be removed.**
>
> Thanks for pointing them out! We have corrected them in the revised manuscript.
>
> We hope these responses can address the reviewer’s concerns and look forward to receiving feedback regarding any remaining questions.
>
> [1] Xu et al. Self-supervised Graph-level Representation Learning with Local and Global Structure. ICML 2021

---

> > ### Comment · Reviewer_7ey5 · 2022-08-07
> > **Thank you for your response, and I maintain my score: strong accept.**
> >
> > I sincerely appreciate the authors' response, which tackles all my previous concerns/comments. I recapitulate my questions and the authors' answers one by one as follows:
> >
> > ---
> >
> > Q1. Regarding the discussion with the relevant work [1], thank you for explaining the differences between the proposed DSSL and the suggested work [1], which is convincing to me.
> >
> > Q2. Thank you for analyzing how the proposed DSSL can capture long-range dependencies, which makes sense.
> >
> > Q3. Thank you for your clarification in regards to the color in Figure 1.
> >
> > Q4. Thank you for clarifying the intra-cluster variation in Line 172. Your explanation in the response comment makes sense, and I suggest authors carefully include the clarification written in the response comment in the next revision.
> >
> > Q5. Sorry for my misunderstanding in regard to the projector $g_\theta$; I missed the explanation in Line 156.
> >
> > Q6. Thank you for correcting minor typos and page errors.
> >
> > ---
> >
> > Besides the above points, I also have read other reviewers' comments, and I think the authors faithfully address all of them well enough. However, one particular point I would like to note is that, as Reviewer 4zvQ pointed out, vGraph [A] is similar to the proposed DSSL. However, in my understanding, the authors clearly describe which points are different from the suggested related work in terms of motivation, method designs, and theory, and it would be worthwhile to highlight the discussions about such comparisons in the main paper, in the next revision.
> >
> > [A] vGraph: A Generative Model for Joint Community Detection and Node Representation Learning. In NeurIPS 2019.

---

> > > ### Author Response · Authors · 2022-08-07
> > > **Thank you for your comments!**
> > >
> > > Thank you for your detailed and helpful comments, and we are grateful for your approval. We will highlight our discussions about vGraph [1] in the next revision.
> > >
> > > [1] vGraph: A Generative Model for Joint Community Detection and Node Representation Learning. In NeurIPS 2019.

---

### Official Review · Reviewer_4zvQ · 2022-07-12

**Rating:** 7
**Confidence:** 5
**Soundness:** 3 good
**Presentation:** 3 good
**Contribution:** 3 good

**Summary:**

This paper focuses on self-supervised learning for node representation learning on non-homophilous graphs. The authors propose a framework called DSSL. DSSL leverages a probabilistic formalization of the problem, and it introduces a latent variable for modeling the semantic structure of graphs. Experiments on a few graph benchmarks show promising results of DSSL.

**Questions:**

See the weaknesses above.

**Strengths And Weaknesses:**

----- Strengths:

(1) The problem is important.
Node representation learning is a fundamental problem in the graph machine learning area for its wide downstream applications. Many recent works propose to use graph neural networks for node representation learning, but these methods require a large amount of labeled nodes for training. Different from these works, this paper focuses on self-supervised learning for node representation learning, which is an important problem.

(2) The proposed DSSL framework is principled.
A framework called DSSL is proposed. The framework uses a probabilistic formalization of the problem, and a mixture model is used to characterize graph structures, where a discrete latent variable is introduced. During training, gumbel-softmax is used to deal with the discrete latent variable. As discussed in the paper, DSSL can be theoretically viewed as to maximize the mutual information between node representations and global semantics. Overall, the DSSL framework is principled with good theoretical results, and I quite like this approach.

(3) The results are encouraging.
In the experiment, the authors use a few graph benchmarks for comparison, and DSSL outperforms many competitive baselines.

----- Weaknesses:

(1) Similar ideas of self-supervised learning have been explored in literature.
In terms of methodology, the major idea of DSSL is to maximize the conditional probability for each edge. Although this is an intuitive idea, the idea has been widely explored by existing studies. For example, classical graph embedding algorithms (e.g., DeepWalk, LINE, node2vec) also try to maximize the conditional probability for model training. Some more recent works (e.g., GraphSAGE) leverages similar ideas. The major difference between DSSL and these works is that DSSL uses a mixture-model formalization to define the conditional probability on each edge, which is able to approximate true conditional probabilities more precisely, but this idea has also been touched by existing works (e.g., vGraph: A Generative Model for Joint Community Detection and Node Representation Learning). Given all these works, the idea of DSSL is not very new. Also, vGraph is not discussed in the paper, making it hard to judge the contribution of DSSL over vGraph. Thus, it would be helpful if the authors could discuss the advantage of DSSL over vGraph in the revised draft.

(2) Whether non-homophilous graphs are worth studying?
The major focus of the paper is on non-homophilous graphs. However, I feel like non-homophilous graphs are not so popular in practice, and homophily exists in most real graphs. Besides, in terms of modeling, I feel like homophilous graphs and non-homophilous graphs are not so different. For example, if we look at the results in the experiment, we can see GraphCL and BGRL achieve competitive results on most homophilous graphs, although they are not specifically designed for non-homophilous graphs. Furthermore, from my understanding, DSSL can be applied to both non-homophilous and homophilous graphs, and I don't see special designs targeting at dealing with non-homophilous graphs. Therefore, I think what DSSL does is not so consistent with the major claim of the paper, and I wonder why the authors position DSSL as modeling non-homophilous graphs rather than general graphs (i.e., both non-homophilous and homophilous graphs)?

(3) The experiment can be further improved.
In the experiment, the authors compare DSSL against several competitive baseline methods. Overall, DSSL outperforms all the methods on non-homophilous graphs. However, the experiment mainly presents quantitative results, showing DSSL is a better model for non-homophilous graphs, but the underlying reason is not well explained and validated. Therefore, it might be more convincing if the authors could do some ablations studies or case studies to show the intuition why DSSL has better performance on non-homophilous graphs.

---

> ### Author Response · Authors · 2022-08-02
> **Response to Reviewer 4zvQ - Part 1**
>
> We are grateful for your approval and sincerely thank you for your time and thoughtful feedback. Please find the following responses to the specific comments.
>
> **Q1. It would be helpful if the authors could discuss the advantage of DSSL over vGraph [1] in the revised draft.**
>
> We thank the reviewer for raising the related work vGraph. We agree with the reviewer that vGraph considers a mixture process to define the conditional probability on each edge and share some similarities with our work. Here, we would like to discuss vGraph and show that our work has several key innovations compared with vGraph from different perspectives. We also added the following discussion in Appendix G.  Please refer to the revised draft for the update.
>
>
> **Motivation**. Although vGraph also considered a mixture process, our motivation is different from vGraph. vGraph assumes that each node $v$ can be represented as a mixture of multiple communities and each community $z$ is modeled as a distribution over the nodes. This motivation results in a generative model for each edge $(u,v)$: $p(u,v)=\int_{z} p(v|z)p(z|u) dz$. However, our motivation is that each node has latent heterogeneous factors which are utilized to make connections to its different neighbors, and the factor denotes various reasons behind why two nodes are connected. Our motivation results in a different generative model: $p(u,v)=\int_{z} p(v|z,u)p(u|z)p(z)dz$. Due to the differences in motivation, our learning objective and framework are also inherently different from vGraph. Specifically, our loss encourages the model to reconstruct the local neighbors while considering different semantic shifts captured by latent variables (see equation (5)), which is important for non-homophilous graphs and different from the learning objective of vGraph.
>
> **Framework**. vGraph is a classical shallow network embedding algorithm. However, we instead tackle the problem of self-supervised learning with GNN. Compared to the shallow method vGraph, our work is flexible to different GNN encoders and can easily incorporate node attributes with message passing and can be fine-tuned to different tasks. More importantly, conducting self-supervised learning with GNN brings some unique optimization challenges, such as trivia and degenerated solutions as shown in section 3.4. We propose a novel self-supervised learning framework (see Figure 6) to address these challenges, which are also inherently different from vGraph.
>
> **Theory**. As also noticed by the reviewer, we provide a theoretical analysis to show the connection between our objective and the mutual information maximization and prove that the learned representations by our objective provably enjoy good downstream performance. This theoretical analysis has not been proposed by vGraph and further strengthens our contribution over vGraph.
>
> Given the above, we believe that our work offers a novel perspective and provides a novel framework with extensive analysis and our contributions are significant given vGraph. But we agree with the reviewer that we share some similarities with vGraph. Thus, we find it might be a good idea to empirically compare our DSSL with vGraph. We run additional experiments on vGraph with the code provided by the authors. We have included the results and more analysis in Appendix F.3. We can find that our DSSL achieves the better performance compared to vGraph on the large-scale non-homophilous graph.
>
> [1] vGraph: A Generative Model for Joint Community Detection and Node Representation Learning. In NeurIPS 2019.

---

> > ### Author Response · Authors · 2022-08-02
> > **Response to Reviewer 4zvQ - Part 2**
> >
> > **Q2. Whether non-homophilous graphs are worth studying? and why the authors position DSSL as modeling non-homophilous graphs rather than general graphs?**
> >
> > Thanks for your very insightful question!
> >
> > Firstly, we would like to clarify that non-homophilous graphs are indeed worth studying. Actually, there are many real-world graphs exhibiting non-homophilous properties. For instance, in online transaction networks, fraudsters are more likely to build connections with customers rather than other fraudsters [1]; in molecular networks, protein structures are more likely composed of different types of amino acids that are linked together [2, 3]. Recent work [4] also released high-quality datasets covering different non-homophilous real-world applications. Thus, studying the non-homophilous graph is indeed an important research problem.
> >
> > Secondly, we want to explain that there are actually no "absolute" homophilous or non-homophilous graphs in the real world. Currently, in literature, researchers utilize the homophily ratio, defined as the fraction of edges connecting nodes with the same labels, to determine whether a graph is homophilous or non-homophilous, graphs with higher homophily ratios are homophilous, and those with lower ratios are non-homophilous. Thus, non-homophilous graphs do not mean there are no non-homophilous edges. For instance, there are 30\% homophilous edges and 54.5\% homophilous edges in the widely used non-homophilous graph dataset Cornell and Twitch, respectively. We provide the statistics of other datasets in Table 4. This phenomenon can explain why those GraphCL and BGRL can achieve competitive results on some non-homophilous graphs.
> >
> >
> > As we mentioned above, the  "homophilous" or "non-homophilous" is determined only based on the global summary statistics, which capture the average mixing patterns in the graph. Thus, non-homophilous graphs may still have homophilous edges. As we discussed in the introduction, real-world graphs may show heterogeneous and diverse mixing patterns wherein certain parts of the graph are homophilous while others are non-homophilous. Thus we design a learning strategy to capture these diverse mixing patterns. Specifically, our DSSL utilizes the latent variables to decouple the heterogeneous and diverse patterns in neighbors.
> >
> > The core idea of our DSSL is to decouple the heterogeneous and diverse patterns in neighbor distributions, and the diverse patterns should occur much more frequently in non-homophilous graphs compared to homophilous graphs; thus, we position DSSL as modeling non-homophilous graphs. For highly homophilous graphs such as Cora, which may also have a small amount of heterophilous edge, explicitly modeling and decoupling neighbor distribution may not benefit significantly. Thus, common self-supervised learning can achieve very competitive performance, as shown in our experiments. We will clearly state this in the main text of our version.
> >
> >
> > **Q3. Ablations studies or case studies to show the intuition why DSSL has better performance on non-homophilous graphs.**
> >
> > Thanks for your good suggestions. We have conducted extensive ablations studies and case studies showing the intuition of DSSL. Specifically, we have conducted ablations studies in Tables 2 and 10. For case studies, in Figure 5 (b), we show that the latent distribution leaned by DSSL shares a similar pattern with the cross-class neighborhood similarity. The case study experiments in Figure 5 (c) and (d) and Figure 10 show that, in many cases, the learned latent factors share a similar distribution for the same type of link and different types of links exhibit different latent distributions. This interprets the reason why capturing the learned semantic shift on the local loss can improve performance. We also have included a new case study experiment to show how our learned global semantic clusters perform on the long-range node in Appendix F.7 in our revision. Please see our revision for the update.
> >
> > We hope we have addressed your concerns.  If you have any further concerns or questions, please do not hesitate to let us know. We will respond to them timely.
> >
> > [1] NetProbe: A Fast and Scalable System for Fraud Detection in Online Auction Networks. In WWW 2007.
> >
> > [2] Beyond Homophily in Graph Neural Networks: Current Limitations and Effective Designs. In NeurIPS 2020.
> >
> > [3] Graph Neural Networks for Graphs with Heterophily: A Survey. arXiv preprint arXiv:2202.07082.
> >
> > [4] Large Scale Learning on Non-Homophilous Graphs: New Benchmarks and Strong Simple Methods. In NeurIPS 2021.

---

> > > ### Comment · Reviewer_4zvQ · 2022-08-10
> > > **Response**
> > >
> > > Thanks the authors for the detailed feedback, which addressed most of my concerns. I have raised my rating accordingly.

---

### Official Review · Reviewer_ANod · 2022-07-12

**Rating:** 5
**Confidence:** 5
**Soundness:** 3 good
**Presentation:** 3 good
**Contribution:** 2 fair

**Summary:**

The paper proposes a novel self-supervised learning approach for node representation learning on non-homophilous graphs. The proposed approach model the neighbors of a central node via a generative process. In addition to learning the patterns of local neighbors in the graph, the proposed approach also learns global semantic information in the node representations with parameterized prototypes. On top of that, the proposed method seeks to find different underlying semantics in different patterns of local neighbors in the graph.

**Questions:**

1. Any reasons of using 60% 20% 20% data splits rather than standard splits as used in other papers?
2. It would be great if the author could compare the proposed method against works with similar efforts, as stated in Weakness 1.
3. Could the authors provide more details on the complexity analysis?



**Strengths And Weaknesses:**

Strengths

1. This paper presents a framework for self-supervised learning on non-homophilous graphs, which is meaningful and well-motivated.
2. The paper shows a reasonable amount of experimental results on data of different homophily levels.
3. The paper performs thorough ablation studies on different components of the model.

Weaknesses
1. Compared graph self-supervised learning methods are not comprehensive. For self-supervise graph learning methods, the performance of included baselines such as MVGRL and BGRL are not SOTA. It is better to include methods such as [1][2][3] which has much better performance. Also, the proposed method aims to learn the local structure and the global semantic information with self-supervised learning. Works with similar motivations have been proposed in the community. For example, [3] also proposes jointly considering structural and neighbor information for self-supervised learning. [4] also proposes a clustering objective. [5] also uses the idea of prototypes to capture the global semantic knowledge

2. For the experiments, the authors didn't use standard data split protocol (20-shot per class). Using 60% training data is somehow contradict with the goal of self-supervised learning and it is hard to fairly compare with the reported results in our papers. Meanwhile, considering that the proposed method consists of multiple components, ablation studies are vital to validate the power of semantic shift and personalized posterior. It would be great if the author can perform the ablation studies on multiple non-homophilous graph datasets and even on homophilous graph datasets.

3. Minor issues
Line 39 constructs -> construct

Line 96 have achieve -> have achieved

Line 182 equivalent to maximize -> equivalent to maximizing

Line 164 relation context -> relational context

Line 220 undated -> updated

Line 250 affect -> affects

Line 303 an relative improvement -> a relative improvement

[1] Jiao et al. "Sub-graph Contrast for Scalable Self-Supervised Graph Representation Learning", ICDM 2020

[2] Jin et al. "Multi-Scale Contrastive Siamese Networks for Self-Supervised Graph Representation Learning", IJCAI 2021

[3] Mo et al. "Simple unsupervised graph representation learning", AAAI, 2022

[4] Zhao, Han, et al. "Graph Debiased Contrastive Learning with Joint Representation Clustering" IJCAI 2021

[5] Ding et al. "Structural and Semantic Contrastive Learning for Unsupervised Node Representation Learning", arXiv 2022

---

> ### Author Response · Authors · 2022-08-02
> **Response to Reviewer ANod - Part 1**
>
> We thank the reviewer for the detailed comments and suggestions, which are helpful for the improvement of the paper. The following is our point-to-point response to the reviewer’s concerns and comments:
>
> **Q1. Works with similar motivations have been proposed in the community [3,4,5].**
>
> Thanks for your insightful question. We also thank the reviewer for pointing out the missing related works [3,4,5], which are important references for our paper. However, our work indeed differs from these works in a number of important ways:
>
> **Motivation**. Our motivation is inherently different from works [3, 4, 5]. Specifically, [3] jointly considers structural information and neighbor information to explore their complementary information for better performance, [4, 5] consider adding a clustering layer or a prototype clustering component to capture the global information. Although there is one component in our loss that also encourages capturing global information, our motivation and contribution are more than that. Actually, the main motivation of our work is that the neighborhood distributions on non-homophilous graphs typically follow heterogeneous and diverse patterns. Thus, the unique challenge is that one needs to decouple the unobserved latent variables in these diverse patterns for better node representations. Given this, we propose a framework to model this semantic shift between neighbors via a generative process. This kind of motivation is different from and has never been discussed by [3, 4, 5].
>
>
> **Method**. The works [3,4,5] are all mainly built on the principle of contrastive learning. However, our work is a generative model without relying on the negative sampling required by contrastive learning. In addition, different from [3,4,5], we develop a novel scalable training algorithm with variational inference in Appendix 3.3 and propose an effective optimization strategy with the graph neural network in Appendix 3.3. Thus, the technical contribution of our work should be significant and different from [3,4,5].
>
>
> **Theory**. Our DSSL can be theoretically viewed as to maximize the mutual information between node representations and global semantics, and the learned representations by our objective provably enjoy the good downstream performance. Thus our DSSL framework is principled with some novel theoretical results. This also makes our work different from past works [3,4,5] and further strengthens our contribution.
>
> **Experiments** We conduct extensive experiments and analysis on self-supervised learning on non-homophilous graphs, while works [3,4,5] only work on homophilous graphs. For non-homophilous graphs, to date, self-supervised learning receives little attention. Thus, we believe that showing our DSSL works well for non-homophilous graphs is another important contribution.
>
> We again thank the reviewer for pointing out these relevant references that will improve our paper. We have cited these works and added the above comparison in Appendix G in our revision. Please refer to the paper for the update.
>
>
> **Q2. Compare the proposed method against works [1,2,3].**
>
> Thanks for your suggestion! Given the short rebuttal period, we only can include an empirical comparison with [1,2] in Appendix F.4. We will also include the comparison with [3] in our final revision. From Table 8 in Appendix F.4, we can observe that our DSSL could achieve competitive results compared with the most powerful baseline in homophilous graphs and achieves the best performance in several non-homophilous graphs.
>
>
> [1] Sub-graph Contrast for Scalable Self-Supervised Graph Representation Learning, In ICDM 2020.
>
> [2] Multi-Scale Contrastive Siamese Networks for Self-Supervised Graph Representation Learning. IJCAI 2021.
>
> [3] Simple unsupervised graph representation learning. In AAAI 2022.
>
> [4] Graph Debiased Contrastive Learning with Joint Representation Clustering. In IJCAI 2021.
>
> [5] Structural and Semantic Contrastive Learning for Unsupervised Node Representation Learning. In ArXiv 2022.

---

> > ### Author Response · Authors · 2022-08-02
> > **Response to Reviewer ANod - Part 2**
> >
> >  **Q3. Any reasons of using 60\% 20\% 20\% data splits rather than standard splits as used in other papers?**
> >
> > We thank the reviewer for this question. The reason why we utilize the random splitting 60\% 20\% 20\% is that we want to strictly follow previous works for non-homophilous graphs [1, 2, 3, 4], which had used the same random splitting, and to be consistent across all datasets in our experiments. Thus, our splits are consistent with current work, and it is easy to compare our results with those for non-homophilous graphs [1, 2, 3, 4]. Nevertheless, we agree that it might be a good idea to also compare our DSSL with baselines on the sparse semi-supervised splitting (20-shot per class) [5] on those homophilous graphs such as Cora, Citeseer, and PubMed. Thus, we provide the additional results on this setting in Appendix F.4. From these additional results, we can find that our DSSL still can achieve better (or competitive) performance compared to baselines.
> >
> >
> >
> > **Q4. Perform the ablation studies on multiple non-homophilous graph datasets and even on homophilous graph datasets.**
> >
> > Thanks for your suggestion. As also suggested by reviewer mEXA, we can generate some synthetic datasets with only homophilous structure or non-homophilous structure to verify the proposed method. Thus, we also conduct ablation studies on the graph with various homophily ratios in Appendix F.4 to verify the significance of our designed components. Please see our revision for the update.
> >
> >
> >
> > **Q5. Could the authors provide more details on the complexity analysis?**
> >
> > Yes! We have provided a detailed time complexity analysis and empirical running time comparison in Appendix C. Please refer to our revision for this analysis.
> >
> > Shortly, the total time complexity of DSSL  is $\mathcal{O}\left(d L|\mathcal{E}|+Nd^{2} L+C_{\text {predictor}}N+C_{\text {projector}}N+KNd+|\mathcal{E}|d\right)$ with GCN [5] as backbone where $C_{\cdot}$ are constants depending on architecture of different components and $d$ denotes the hidden dimensions.  With the similar notations, we can express the time complexity of GCA as $\mathcal{O}\left(d L|\mathcal{E}|+Nd^{2} L+C_{\text {projector}}N+N^{2}d\right)$ and that of  BGRL  as $\mathcal{O}\left(d L|\mathcal{E}|+Nd^{2} L+C_{\text {predictor}}N+Nd\right)$. We can note that the time complexity of GCA scales quadratically in the size of the nodes, and both BGRL and our DSSL linearly increase with the number of nodes $N$ when $\mathcal{E}$ is proportional to $N$. Thus the overall time complexity of our DSSL is at the same level as BGRL, which is a large-scale self-supervised learning method. We further provide an empirical comparison of the training time per iteration in Table 3 in Appendix C. Our DSSL has a shorter training time than MVGRL and DGI since our DSSL does not require negative examples. Additionally, our DSSL has a similar efficiency compared to BGRL but yields better performance on non-homophilous graphs.
> >
> > **Typos**. We have corrected the typos and thanks for your time and efforts in reviewing our paper. If you have any further concerns or questions, please do not hesitate to let us know. We will respond to them timely.
> >
> >
> > [1] Geom-GCN: Geometric Graph Convolutional Networks. ICLR 2020.
> >
> > [2] Adaptive Universal Generalized PageRank Graph Neural Network. In ICLR 2021.
> >
> > [3] Node Similarity Preserving Graph Convolutional Networks. WSDM 2021.
> >
> > [4] Simple and Deep Graph Convolutional Networks. ICML 2020.
> >
> > [5] Semi-Supervised Classification with Graph Convolutional Networks. ICLR 2017.

---

### Official Review · Reviewer_mEXA · 2022-07-16

**Rating:** 5
**Confidence:** 4
**Soundness:** 2 fair
**Presentation:** 3 good
**Contribution:** 3 good

**Summary:**

This paper proposes a new method, called decoupled self-supervised learning (DSSL), for self-supervised learning of node representation on non-homophilous graphs. Experiments on reals datasets are used to verify the effectiveness of the proposed method.

**Questions:**

1. What is the time complexity of the proposed method? Is the proposed method slower than existing methods? Why not perform experiments on larger datasets?
2. How to explain the phenomenon that existing methods based on homopilous assumption also achieve comparable accuracy with the proposed method?
3. Can we generate some synthetic datasets with only homophilous structure or non-homophilous structure to verify the property of the proposed method and existing methods?


**Limitations:**

The authors have adequately addressed the limitations and potential negative societal impact of their work.

**Strengths And Weaknesses:**

Strengths:
1. It is interesting to study the self-supervised learning problem about node representation learning on non-homophilous graphs, because most existing graph self-supervised learning methods assume homophily in graphs.
2. The proposed method seems to be reasonable.
3. Theoretical analysis is provided.
4. Better accuracy is achieved by the proposed method, compared with existing methods.
5. The paper is well written with good organization.

Weaknesses:
1. Time complexity analysis is not provided. The datasets adopted for evaluation are relatively small. Is it due to high time complexity of the proposed method? As we know, most self-supervised learning methods are proposed for large-scale datasets.
2. On some datasets with non-homophilous structure, like Twitch, existing methods based on homopilous assumption also achieve comparable accuracy with the proposed method, which seems to contradict with the motivation of this paper.

---

> ### Author Response · Authors · 2022-08-02
> **Response to Reviewer mEXA - Part 1**
>
> We thank the reviewer for the detailed comments and the insightful question. We respond to the reviewer’s question mentioned in the comments below:
>
> **Q1. What is the time complexity of the proposed method? Is the proposed method slower than existing methods?**
>
> Thanks for your insightful question. We have provided detailed time complexity analysis and empirical running time comparison in Appendix C. Please refer to our revision for the update.
>
> Shortly, the time complexity of DSSL  is $\mathcal{O}(d L|\mathcal{E}|+Nd^{2} L+(C_{\text {predictor}}+C_{\text {projector}})N+KNd+|\mathcal{E}|d)$  with GCN [1] as the backbone where $C_{\cdot}$ are constants depending on architecture of the different components and $d$ denotes the hidden dimensions.  With the similar notations, we can express the time complexity of GCA [2] as $\mathcal{O}\left(d L|\mathcal{E}|+Nd^{2} L+C_{\text {projector}}N+N^{2}d\right)$ and that of  BGRL [3] as $\mathcal{O}\left(d L|\mathcal{E}|+Nd^{2} L+C_{\text {predictor}}N+Nd\right)$. Typically, the architecture for predictor and projector is one-layer MLP for all methods, thus the $C_{\text {predictor}}$ and $C_{\text {projector}}$ in different methods should be similar. Based on the analysis above, we can note that the time complexity of GCA scales quadratically in the size of the nodes, and both BGRL and our DSSL linearly increase with the number of nodes $N$ when $\mathcal{E}$ is proportional to $N$. Thus the overall time complexity of our DSSL is at the same level of BGRL which is a large scale self-supervised learning method.
>
> We further provide the empirical comparison of the training time per iteration in Table 3 in Appendix C. We can observe that
> our DSSL has a shorter training time than MVGRL [4] and DGI [5] since our DSSL does not require negative examples. In addition, our DSSL has similar efficiency compared to BGRL but yields better performance on non-homophilous graphs.
>
> Given the above, we can believe that the superior performances of DSSL do not benefit from larger time complexity, and DSSL is not slower than existing methods.
>
> **Q2. Experiments on larger datasets.**
>
> Thanks for your suggestions. We want to kindly remind the reviewer we have tested our method on a relatively larger non-homophilous graph Penn94 released by recent work [6] compared to widely-used small non-homophilous datasets such as Squirrel and Cornell. Nevertheless, we agree with the reviewer that it would be better to conduct experiments on much larger datasets. Thus, we conduct additional experiments on a larger non-homophilous dataset, ArXiv-year [6], which has 169,343 nodes and 1,166,243 edges. The results and analysis have been included in Table 7 in Appendix F.3. We can see that our DSSL is also able to achieve the best performance on ArXiv-year compared to several recent self-supervised methods on GNN.
>
>
> **Q3. How to explain the phenomenon that existing methods based on homopilous assumption also achieve comparable accuracy with the proposed method?**
>
> Thanks for your question. The reason is that there are actually no "absolute" homophilous or non-homophilous graphs. Typically, we utilize the homophily ratio, defined as the fraction of edges connecting nodes with the same labels, to determine whether a graph is homophilous or non-homophilous. Graphs with higher homophily ratios are homophilous, and those with lower ratios are non-homophilous. The  "homophilous" or "non-homophilous" is determined based on the global summary statistics. Thus, non-homophilous graphs may still have homophilous edges. For instance, there are 30\% homophilous edges and 54.5\% homophilous edges in the widely used non-homophilous graph datasets Cornell and Twitch, respectively. We provide the statistics of edge homophily of other datasets in Table 4. This phenomenon can explain why existing methods based on homophilous assumptions can achieve comparable accuracy on some non-homophilous graphs such as Cornell and Twitch.
>
> The phenomenon matches our key motivation that real-world graphs show heterogeneous and diverse mixing patterns wherein certain parts of the graph are homophilous while others are non-homophilous, and we need to capture these patterns in the representation learning process effectively. Our motivation is also consistent with the experimental results that our DSSL can generally perform better on various graphs with various homophily ratios.
>
>
>
> [1] Semi-Supervised Classification with Graph Convolutional Networks. In ICLR 2017
>
> [2] Graph Contrastive Learning with Adaptive Augmentation. In WWW 2021.
>
> [3] Large-Scale Representation Learning on Graphs via Bootstrapping. In ICLR 2022.
>
> [4] Contrastive Multi-View Representation Learning
> on Graphs. In ICML 2020.
>
> [5] Deep Graph Infomax. In ICLR 2019.
>
> [6] Large Scale Learning on Non-Homophilous Graphs: New Benchmarks and Strong Simple Methods. In NeurIPS 2021.

---

> > ### Author Response · Authors · 2022-08-02
> > **Response to Reviewer mEXA - Part 2**
> >
> > **Q4. Can we generate some synthetic datasets with only homophilous structure or non-homophilous structure to verify the property of the proposed method and existing methods?**
> >
> > Thanks for your insightful question! Yes, we can generate some synthetic datasets by artificially controlling the homophily levels. Actually, there are some previous works [1, 2, 3] generating different synthetic datasets to evaluate their GNN architectures on semi-supervised settings. Similarly, we can evaluate our DSSL and existing self-supervised learning methods with these synthetic datasets. Specifically, in Appendix F.4, we further provide the additional comparisons, analysis, and ablation study on the synthetic dataset. Please refer to our revised revision for this part. The results show that our DSSL can generally achieve better performance and effectively adapt to all levels of homophily.
> >
> > We hope these results and our responses help to address the reviewer's concerns. Please let us know if you have any further questions, and we will be glad to write a follow-up response.
> >
> >
> > [1] Beyond Homophily in Graph Neural Networks: Current Limitations and Effective Designs. In NeurIPS 2020.
> >
> > [2] Adaptive Universal Generalized PageRank Graph Neural Network. In ICLR 2021.
> >
> > [3] Universal Graph Convolutional Networks. In NeurIPS 2021.

---

> > ### Comment · Reviewer_mEXA · 2022-08-08
> > **issue about title and motivation**
> >
> > We thank the authors for the response.
> >
> > Since the authors state that "there are actually no absolute homophilous or non-homophilous graphs" in the response, I think there exist major issues about the title and motivation of this paper. The title of this paper is about "non-homophilous graphs", and the motivation is to "design an effective self-supervised scheme for node representation learning in non-homophilous graphs". This is also why I ask the authors to generate some synthetic datasets with absolute homophilous or non-homophilous graphs for verification. Although the authors have added some results on synthetic datasets in response, the synthetic datasets in response (appendix) do not satisfy the requirement in my original comment.

---

> > > ### Author Response · Authors · 2022-08-09
> > > **Further Response to Reviewer mEXA**
> > >
> > > We sincerely thank the reviewer for the further comments!  Please see our following clarification for your concerns:
> > >
> > >  **Q1. Questions about the title and motivation.**
> > >
> > > Firstly, we want to apologize for the confusion and unclear writing of a sentence in our rebuttal. In our response, the sentence “there are actually no absolute homophilous or non-homophilous graphs” just means that
> > >
> > > ***in the real-world graph benchmarks [1,2,3,4],  the non-homophilous graphs may still have a little bit homophily edges, and the homophilous graphs may still have a little bit non-homophily edges.*** Thus, the non-homophilous graphs do not mean there are absolutely no homophily edges.
> > >
> > > In the literature [1,2,3,4], the researchers typically categorize graph benchmarks based on the following rule: graphs with higher homophily ratios are homophilous, and those with lower ratios are non-homophilous.   The homophily ratios are typically defined as the fraction of edges connecting nodes with the same labels. In other words,  ***"Homophilous graphs" have a relatively high homophily ratio close to 1, where Cora (0.81), Citeseer (0.74), and PubMed (0.80) are typical examples. "Non-homophilous graphs" such as Texas (0.11), Cornell (0.30), and ArXiv-year (0.22) have relatively low homophily ratios*** [1,2,3,4].
> > >
> > > ***Our title***. Thus, to keep consistent with previous works [1,2,3,4],  the title "non-homophilous graphs" in our work indicts those graphs which have relatively low homophily ratios.
> > >
> > > ***Our motivation***. Since non-homophilous graphs have a relatively low homophily ratio, thus linked nodes often do not belong to the same class. Current graph self-supervised learning methods that rely on the homophily assumption (neighboring nodes have similar representations) cannot perform well on non-homophilous graphs. The motivation of our paper is "design an effective self-supervised scheme for node representation learning in non-homophilous graphs."  Our model has the flexibility to decouple the heterogeneous and diverse patterns in non-homophilous graphs, account for the semantic shift in the local neighborhood, and capture the global semantic dependencies.
> > >
> > >
> > > **Q2. The synthetic datasets in response (appendix) do not satisfy the requirement in my original comment.**
> > >
> > > We made a small ***reference typo*** in our initial response.  ***In Appendix F. 5 (not Appendix F.4)***, we have provided the additional comparisons, analysis, and ablation study on the synthetic dataset. Please refer to our revised revision for this part. The results in Tables 10 and 11  show that our DSSL can generally achieve better performance and effectively adapt to all levels of homophily (h = [0.0, 0.25, 0.50, 0.75, 1.0]) where $h=1$ indicts "absolute" homophilous graph (no non-homophilous edges) and $h = 0$ indicts "absolute" non-homophilous graph (no homophily edges). For better reference, we also provide the results in Table 10 here:
> > >
> > > | Homophily ratio h  | &nbsp; &nbsp;  0.00  | &nbsp; &nbsp; 0.25 | &nbsp; &nbsp; 0.50 | &nbsp; &nbsp; 0.75| &nbsp; &nbsp; 1.0 |
> > > |--- | --- | --- | ---| ---| ---|
> > > |GAE | 18.28±0.52 |  25.49±0.37  | 50.62±0.28 | 82.05±0.38 | **95.27±0.11** |
> > > |DGI |  40.21±0.43  |  42.38±0.26   | 57.31±0.36 | 85.21±0.43 | 92.12±0.43 |
> > > |MVGRL |  42.69±0.32  |  44.35±0.40  | 59.43±0.27 | 86.65±0.39 | 93.11±0.35 |
> > > |BGRL |  45.31±0.47  |  47.52±0.38  | 61.37±0.26 | 86.21±0.24 | 93.28±0.37 |
> > > |DSSL |  **49.19±0.62**  |  **51.35±0.51**  | **64.78±0.25** | **87.59±0.46** | 94.83±0.55 |
> > >
> > >
> > >
> > > We believe this synthetic dataset matches the reviewer's original comment: "Can we generate some synthetic datasets with only homophilous structure or non-homophilous structure."  If not, could the reviewer provide more details about the requirements?
> > >
> > >
> > > We thank the reviewer for providing valuable and thoughtful comments on our paper!  We hope that our responses have fully addressed all of the reviewer’s concerns and remain committed to clarifying any further questions that may arise during the discussion period.
> > >
> > >
> > > [1] Large Scale Learning on Non-Homophilous Graphs: New Benchmarks and Strong Simple Methods. In NeurIPS 2021.
> > >
> > > [2] Beyond Homophily in Graph Neural Networks: Current Limitations and Effective Designs. In NeurIPS 2020.
> > >
> > > [3] Graph Neural Networks for Graphs with Heterophily: A Survey. arXiv preprint arXiv:2202.07082.
> > >
> > > [4] Geom-GCN: Geometric Graph Convolutional Networks. In ICLR 2020

---

> > > > ### Comment · Reviewer_mEXA · 2022-08-10
> > > > **explanation of new result**
> > > >
> > > > From the new results, can we get the conclusion that both models designed for homophilous structure and models designed for non-homophilous structure can achieve good performance on homophilous graphs but bad performance on non-homophilous graphs?

---

> > > > > ### Author Response · Authors · 2022-08-10
> > > > > **Further Response to Reviewer mEXA about the Explanation of New Results**
> > > > >
> > > > > We appreciate the reviewer's further insightful comments. We respond to the comments as follows.
> > > > >
> > > > > **Firstly, we would like to say that conducting self-supervised learning on non-homophilous graphs is much more challenging compared to homophilous graphs.**
> > > > >
> > > > > In homophilous graphs, the linked nodes typically belong to the same class labels. Thus, we can directly assume that neighboring nodes have similar representations (homophily assumption) and include this as inductive bias in the self-supervised learning process. Instead, in non-homophilous graphs, linked nodes have different class labels. The neighborhood class distributions typically follow much more heterogeneous and diverse patterns. Thus, it is much more challenging to design self-supervised learning for non-homophilous graphs.
> > > > >
> > > > > The idea of our DSSL is trying to explicitly decouple the heterogeneous and diverse patterns in local neighborhood distributions in non-homophilous graphs. Specifically, we model the generation of neighbors by assuming each node has latent heterogeneous factors utilized to make connections to its different class neighbors. Thus, our assumption is more general than the standard homophily assumption. **This is why our algorithm can consistently achieve better performance in non-homophilous graphs and competitive performance on homophilous graphs.**
> > > > >
> > > > >
> > > > > Secondly, since there is no way to access the latent patterns of neighborhoods directly, the lack of supervision prevents us from **perfectly** decoupling the distribution of neighborhoods.
> > > > >
> > > > > **Thus, although our algorithm can consistently achieve better performance than other baselines in non-homophilous graphs, it achieves inferior performance on non-homophilous graphs compared to homophilous graphs since learning with non-homophilous graphs is more challenging than homophilous graphs.**
> > > > >
> > > > > **Another possible reason why all baselines, including our DSSL, achieve inferior performance on non-homophilous graphs than on homophilous graphs is that we use GCN as the backbone for all self-supervised learning methods in the synthetic dataset.**
> > > > >
> > > > > Many recent works [1, 2, 3, 4, 5, 6]  have shown that GCN cannot perform very well on non-homophilous graphs compared to homophilous graphs. Thus, utilizing other advanced GNN architectures [1, 2, 4, 5, 6] with our algorithm may further improve self-supervised learning performance on non-homophilous graphs. Since our work mainly focuses on designing the self-supervised algorithm, not the specific GNN architectures, we leave this as future work. We believe applying our self-supervised learning algorithm to other advanced GNN architectures is an interesting future work direction.
> > > > >
> > > > > We thank the reviewer for the comments and questions. In light of these responses, we hope we have addressed your questions and sincerely hope you consider raising your score. If you have any other question/concern unaddressed, please share with us and we will be glad to attend these points.
> > > > >
> > > > >
> > > > >
> > > > > [1] Beyond Homophily in Graph Neural Networks: Current Limitations and Effective Designs. In NeurIPS 2020.
> > > > >
> > > > > [2] Large Scale Learning on Non-Homophilous Graphs: New Benchmarks and Strong Simple Methods. In NeurIPS 2021.
> > > > >
> > > > > [3] Graph Neural Networks for Graphs with Heterophily: A Survey. arXiv preprint  arXiv:2202.07082.
> > > > >
> > > > > [4] Adaptive Universal Generalized PageRank Graph Neural Network. In ICLR 2021.
> > > > >
> > > > > [5] Node Similarity Preserving Graph Convolutional Networks. In WSDM 2021.
> > > > >
> > > > > [6] Simple and Deep Graph Convolutional Networks. In ICML 2020

---

### Meta-Review · Area_Chair_zJjD · 2022-08-27

**Recommendation:** Accept
**Confidence:** Certain

**Metareview:**

This paper proposes a self-supervised learning method for graph-structured data, which can better explore global and local semantic dependencies. The method uses a probabilistic framework based on the mutual information maximization principle. Theoretical analysis is also developed to show a tighter upper bound on the downstream Bayes error can be obtained by the proposed method. After the rebuttal, all reviewers generally appreciate contributions made by this submission.

However, it is suggested to revise the following problem in the final version.
- As identified by the first three reviewers, this paper does not specifically target at non-homophilous graphs. Instead it seems to can better capture semantic information on graphs on a more non-local level as emphasized by the last reviewer. The connections between global semantic and non-homophilous graphs need to be clarified.
- The above problems make reviewers wonder the necessity of study non-homophilous graphs, whether sufficient methods are compared in homophilous graphs, and where enough ablation studies are performed.
- While authors also offer theoretical analysis, it is also not clear how obtained results are connected with non-homophilous graphs.

**Award:**

No

---

### Decision · Program_Chairs · 2022-09-14

Accept